# Effect of Homochirality of Dipeptide to Polymers’ Degradation

**DOI:** 10.3390/polym12092164

**Published:** 2020-09-22

**Authors:** Xinqiang Xu, Fuyan He, Wenke Yang, Jinshui Yao

**Affiliations:** 1School of Materials Science & Engineering, Qilu University of Technology (Shandong Academy of Sciences), Jinan 250353, China; xu_xin_qiang@163.com (X.X.); hefuyan555@163.com (F.H.); wkyang@qlu.edu.cn (W.Y.); 2Shandong Provincial Key Laboratory of Processing & Testing Technology of Glass and Functional Ceramics, Jinan 250353, China

**Keywords:** chiral amino acids, dipeptide, polyamide-imides, biodegradability

## Abstract

As natural polymer materials, proteins are readily biodegradable, interestingly, the synthetic polyamides (PAs) that are based on the same amide bonds (also called peptide bonds in proteins) are barely degradable. Whether did the chirality and configuration of the amino acids play an important role. By using different configuration of amino acids, 4 types of polyamide-imides (PAIs) containing dipeptides of LL, DL, LD, and DD configurations, respectively, were synthesized. It was found that the PAIs based on natural LL configuration of dipeptide structure are much more readily biodegradable than those based on non-natural LD, DL, and DD configuration of dipeptides. It was confirmed that the natural L-configuration of amino acids play a critical role in degradability of proteins. And it also suggested that different type and amount of peptide fragments can be introduced in polymer to create series of polymer materials that can be biodegraded at controllable speed.

## 1. Introduction

With the growing problem of white pollution, degradable polymer materials are attracting more and more attention because polymer materials that can be degraded at controllable speed could be used in every kind of fields [1,2,3,4,5,6]. So, the study of influence factors to degradability and degradable speed of polymer materials has important scientific significance. As natural polymer materials, proteins are both biocompatible and readily biodegradable; interestingly, the synthetic polyamides (PAs) that are based on the same amide bonds (also called peptide bonds in proteins) are barely degradable. It is well known that proteins are formed using chiral amino acids as monomers, with the only exception of glycine that is achiral; so, chemically, proteins are PAs, while monomers of synthesized polyamides are achiral amino acids or diacids and diamines [7,8]. Such differences suggest that the chirality of nature may play an essential role in biodegradation of polymers [9]. In fact, chirality plays an important role in life phenomena [10,11]. But we all know that the proteins and peptides are all based on L-amino acids. We have not found D-amino acids in natural proteins and peptides. Whether or not the configuration plays a decisive role, except for the role of chirality, remains a question. For this paper, we demonstrated that polymers based on natural LL configuration of dipeptide structure are much more readily biodegradable compared to those based on non-natural DD configuration of dipeptides. This result explains the huge difference in biodegradation between proteins and synthesized polyamide-imides (PAIs), and L-configuration amino acids are the natural selected results. It also suggests that a different type and amount of peptide fragments can be introduced in polymer chains to create series of polymer materials that can be biodegraded at controllable speed.

Synthesis and performance evaluation of chiral PAIs based on amino acids have become a focus of recent research, for their high performance, like polyimides, and high processability, like PAs [12,13,14,15,16,17,18]. Furthermore, the degradation properties of PAIs are improved due to the introduction of natural chiral sections, which is obviously of great significance for development of high-performance, high-processability, and biodegradable polymer materials [19,20,21]. However, most of the amino acid-based chiral polymers so far used natural L-α-amino acid-based products as monomers and always introduced independent amide bonds in the main chain of the chiral polymers [19,22]. Studies on peptide-containing chiral polymers have not been reported. It is obvious that introduction of peptides into a polymer would lead to a structure closer to that of natural proteins. It has been proven that polymers containing L-amino acids are much more biocompatible than those containing D-amino acids [23,24]. So, replacing all or some of the natural L-amino acids with D-amino acids, and then comparing their differences in degradability, would gain important insights in factors affecting degradability of polymers. Polymer materials were either degraded as grounded powder in phosphate buffered solution (PBS) that simulates human body environment or as membrane in aqueous soil suspension that simulates soil degradation [25,26,27].

## 2. Materials and Methods

### 2.1. Materials 

L (D)-alanine (Ala), L (D)-phenylalanine (Phe), 4,4′-diaminodiphenyl ether (ODA), triphenyl phosphite (TPP), pyromellitic dianhydride (PMDA), N-methyl-2-pyrrolidone (NMP), pyridine (Py), and dimethyl sulfoxide (DMSO) were obtained from Shanghai Aladdin Biochemical Technology Co. Ltd. (Shanghai, China). Anhydrous calcium chloride (CaCl_2_), thionyl chloride, ethyl acetate, N,N-dimethylformamide (DMF), and anhydrous ethanol were purchased from Tian in Fuyu Fine Chemical Co. Ltd. (Tianjin, China). All chemicals were used directly without further purification.

### 2.2. Characterizations

Powders were mixed with KBr powders, and the mixture was pressed into a tablet. The Fourier transform infrared (FTIR) spectra of samples were recorded using a Nicolet IS10 (New Castle, DE, USA) spectrophotometer. Scanning electron microscopy (SEM) images were collected on a Quanta 200 (FEI, Hillsboro, OR, USA) environmental scanning electron microscope. Proton nuclear magnetic resonance (^1^H NMR) was performed on a Bruker AVANCE II 400 MHz spectrometer (Bruker, Billerica, MA, USA) in DMSO-d_6_. Specific rotations were executed under a WZZ-2 polarimeter (CSOIF Co., Ltd., Shanghai, China) at a concentration of 0.1 g/100 mL in DMSO at 25 °C. Inherent viscosities were measured under a standard procedure using a Technico Regd Trade Mark Viscometer (Cannon, Butler County, PA, USA). Elemental analyses were carried out via a Vario EL III model element analyzer (Elementar, Hamburg, Germany). Thermogravimetric analysis (TGA) on polymers was conducted on a TGA/SDTA851 System (Setaram, Caluire-et-Cuire, France) under nitrogen (N_2_) atmosphere at a flow rate of 10 °C/min. The glass transition temperatures (*T*_g_) of the polymers were studied on a differential scanning calorimetry (DSC, Q-100, TA instruments, New Castle, DE, USA) instrument. The molecular weights of polymers were analyzed with gel permeation chromatography (GPC) and multi-angle laser light scattering (Dawn Heleos, Wyatt, Goleta, CA, USA), using a linear MZGel SD Plus GPC column set (two columns, 5 µm particles, 300 × 8 mm) with DMF as eluent at room temperature with a flow rate of 1 mL/min and a concentration of the polymer of ca. 1 mg/mL.

### 2.3. Synthesis of Monomer 1–4

*N*,*N*′-(pyromellitoyl)-bis-L-Ala diacid chloride and *N*,*N*′-(pyromellitoyl)-bis-D-Ala diacid chloride were prepared by following the analogous method reported in the literature [28].

L-Phe (0.991 g, 6 mmol) and 0.2 mol/L of NaOH solution (30 mL) were added to 100 mL of three-necked round-bottom flask, N,N′-(pyromellitoyl)-bis-L-Ala diacid chloride (0.993 g, 2.5 mmol) was added in when the system was cooled to 0 °C. 2 mol/L of NaOH solution was used to maintain the pH value of the reaction solution at 8–10, and the mixture was reacted for 1 h. Then, the reaction mixture was acidified with 6 mol/L of hydrochloric acid solution until the pH value reached 1–2, and the reaction mixture was extracted with ethyl acetate (4 × 50 mL). The organic layers were combined, dried over anhydrous Mg_2_SO_4_, and concentrated under vacuum to yield the yellow solid.

The above product was dissolved in 30 mL of acetic acid in a 50 mL of round-bottom flask. The mixture was refluxed for 5 h. The solvents were removed under reduced pressure, and 5 mL of cold concentrated hydrochloric acid solution was added. A precipitate was formed, filtered off, washed with water, and concentrated under vacuum to yield the crude product, which was further purified by flash chromatography on silica gel using ethyl acetate/petroleum ether/acetic acid (9:2:2) as the eluent to afford monomer 1 containing L-Ala-L-Phe dipeptide as a white solid. The other three monomers were synthesized in the same way (illustrated in Scheme 1). The structure of the 4 types of monomers was confirmed by ^1^H NMR data, as shown in Appendix A.

### 2.4. Synthesis of PAIs Containing Dipeptide Structure

A solution of monomer 1 (1.309 g, 2 mmol), ODA (0.400 g, 2 mmol), CaCl_2_ (0.500 g), TPP (2.4 mL), and Py (1 mL) in NMP (8.0 mL) was stirred at 120 °C for 8 h. After cooled to room temperature, the resulted polymer was precipitated into ethanol (500 mL). The polymer was collected, washed with ethanol, and dried under vacuum to give PAI 1 containing L-Ala-L-Phe dipeptide (yield 93%, intrinsic viscosity 0.44 dL/g, specific rotation −161.11°.). PAI 2–4 were prepared under this analogous procedure (Scheme 2). Synthesis and some physical properties of polymers are shown in Table 1.

### 2.5. Degradation Experiments of PAI Powders in Phosphate Buffer Solution

PAI powders were crushed into evenly sized particles in an agate mortar. Fifteen hundredths of a gram of PAI was separately poured into 35 mL of PBS (pH = 7.4) for the biodegradation tests in the oscillation incubator at 37 °C with a spinning rate of 160 rpm. It is worth mentioning that the PBS buffer, polymer samples, and glassware were not sterilized before degradation. The biodegraded samples were individually obtained at the end of the 7th week. The degraded products that remained in the supernatant were gained after centrifugation of the above-collected buffer solutions. The received precipitates were washed several times repeatedly with distilled water and dried in a vacuum oven for 48 h for complete drying. The weight losses in percentage for each PAI samples were subsequently recorded as the weight difference between the dried collected PAI samples and the initial ones. The structure changes of biodegraded samples and the degraded products were characterized by means of FTIR.

### 2.6. Degradation of PAI Films in Soil

Thirty-six hundredths of a gram of PAI product was dissolved in 5 mL of DMF under sufficient stirring. Then, the solution was cast on a dry Teflon mold with a diameter of 5 cm, followed by an intensive vacuum drying to yield a final yellow transparent PAI film.

Films of PAI 1–4 were separately immersed into soil suspension (35 mL) for the biodegradation experiments in the oscillation incubator at 37 °C with a spinning rate of 160 rpm. The biodegraded PAI films were separately collected at the end of the 4th week, washed with ethanol, and dried thoroughly in a vacuum oven for 48 h. The morphological changes of the films were observed by means of SEM.

## 3. Results and Discussion

By using amino acids of different configuration, we synthesized 4 types of PAIs containing dipeptides of LL, DL, LD, and DD configurations, respectively (Scheme 2). Chemical structure characterizations of obtained PAI 1–4 were performed by means of FTIR, ^1^H NMR, and elemental analysis. The ^1^H NMR spectrum of PAI 1 confirmed its chemical structure, as shown in Figure 1. The 4 types of polymers showed no discernable difference in ^1^H NMR spectrum. Table 1 shows the good agreement between the calculated and theoretical values on elemental analyses of PAI 1–4, which furthermore proves their correct structures.

The thermal stabilities of PAIs were investigated by TGA and DSC under N_2_ atmosphere at a heating rate of 10 °C/min. The TGA and DSC measurements of PAI 1–4 were explored, as shown in Appendix A, respectively. The *T*_g_ obtained from DSC and the temperatures corresponding to weight loss of 5% (*T*_5_) and 10% (*T*_10_) of PAIs acquired from TGA are summarized in Table 2. The *T*_g_ value of PAI 2 and PAI 3 is lower than the others. The reason for this is that the configuration of the two amino acids in each repeat unit is the opposite. The configuration of the pendant groups on the amino acid is a distribution of up and down, increasing the distance between the molecules and reducing the probability of amide bonds forming hydrogen bonds [29,30].

The temperatures of initial decomposition (*T*_5_ and *T*_10_) which ranged from 272 °C to 290 °C and from 312 °C to 318 °C, respectively, and the high char yields (48.20–52.35%) at 700 °C of PAI 1–4 showed good thermal stabilities of these polymers. The flame-retardant properties of these polymers were evaluated by measuring their limiting oxygen index (LOI) in accordance with Van Krevelen and Hoftyzer equation [31]: LOI = 17.5 + 0.4CR (CR = Char Yield).(1)

Table 2 shows all PAIs showed LOI data ranged from 36.78 to 38.44. Such macromolecules can be classified as self-extinguishing polymers and are considered to be flame retardant (LOI > 28) [32].

After 7 weeks of degradation in PBS at 37 °C, the GPC diagrams of the polymer powders are shown in Appendix A, and the weight loss and molecular weights reduction values of the 4 types of polymer powders are listed in Table 3.

All 4 types of polymers showed significant weight loss and molecular weights reduction, which indicates that the polymers were degraded under the action of microorganisms. However, weight and molecular weights reduction differs among the groups; the largest reduction is observed in PAI 1, the smallest in PAI 4, and PAI 2 and PAI 3 are in-between with similar values. Weight loss and molecular weights reduction value of PAI 1 that contains LL dipeptide is over 5 times that of PAI 4 that contains DD dipeptide, and weight loss and molecular weights reduction value of LD and DL peptide-containing PAI 2 and PAI 3 is also over 3 times that of PAI 4. These results suggest that polymers of different optical activity degrade at significantly different rates. Microorganisms preferentially attack PAI containing dipeptide of LL configuration, while PAI containing dipeptide of DD configuration is not favored by microorganisms.

The 4 types of polymers showed no discernable difference in infrared spectrum before degradation, but certain differences can be observed after degradation (Figure 2).

As shown in Figure 2, infrared spectrum of the residue polymer showed some obvious change: the ether-bond peak (1050 cm^−1^) becomes more apparent, while other peaks, especially the carbonyl peaks (1550 cm^−1^–1780 cm^−1^) associated with peptide and imide groups, weakened. PAI 1 that has the largest weight loss showed only the ether-bond peak and barely any others, which also indicated significant degradation. According to our previous studies on PAIs containing a single amino acid, the difference in amino acid chirality has only a small effect on the degradation of the polymers [22]. But the configuration of amino acids has a great influence on polymers degradation after introducing peptide bonds. Therefore, the breakage of peptide bonds in the degradation process of PAI is dominant, and other groups are secondary.

As for the degradation assay of the four polymers in soil, PAI 1–3 had bacterial colony forming on the surface since week 4 of the experiment, which grew in both size and number in later days. SEM images of the polymers are shown in Figure 3. Compared to PAI 1–3 containing L-amino acids, the surface of PAI 4 showed little changes, with barely any bacterial colonies, except some small surface damages, possibly due to basal hydrolysis [33]. Compared to PAI 2–3 containing LD and DL-dipeptides, the surface of PAI 1 containing LL-dipeptides showed greater changes, with bigger and more bacterial colonies. There are also some differences between the surface of PAI 2 and PAI 3, possibly due to selectivity to every kind of bacteria between L-Ala and L-Phe.

## 4. Conclusions

In conclusion, the natural L-amino acids are more biodegradable compared to D-amino acids, which is the result of natural selection, so that natural biological materials, like proteins, are both biocompatible in living organism and biodegradable after discarded. That is to say, there are two runners (L and D amino acids) in the effect of homochirality of dipeptide to polymers’ degradation, and L-amino acids-based dipeptides won. This is of great guiding significance for development polymer materials that can degrade at controllable speed, i.e., by introducing different types and amount of amino acids or peptides to achieve different degradation speed for different applications.

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
