# Peer review of "Effect of Homochirality of Dipeptide to Polymers’ Degradation"

_polymers, 2020, doi:10.3390/polym12092164_

Round 1

Reviewer 1 Report

Four polymers have been prepared incorporating an Ala-Phe dipeptide of the LL, LD, DL and DD configuration. All polymer samples have been extensively characterized and then subjected to degradation under two different conditions: as a powder suspended in a phosphate buffer and as a membrane in an aqueous soil suspension. In the both above cases the links with the LL-dipeptide were found to decompose significantly faster than those with the DD-enantiomers.

  1. According to stereochemical rules, the LL-polymer must be identical to the DD-analogue with the exception of contacts with living organisms. The degradation experiments of the above four polyamides are exactly the case. I find that this condition is not shown clearly enough in the manuscript. Thus, the fact that the phosphate buffer and the polymer samples have not been previously subjected to sterilization must be clearly spelled out; so that the unavoidable microbial or fungal contamination of the system within the 7 weeks of the experiment presents the only real cause of the difference in the degradation of LL- and DD-enantiomers in the buffer solution. (The pH value of the buffer must be specified).
  2. The signs of the specific optical activity of the LL- and DD-polymer solutions MUST BE OPPOSITE, while in the Table 2 the both are shown to be positive. If I assume that this error is a simple misprint, the difference in the absolute values still appears to be exceedingly big: 153 and 115. These values must be re-examined or this point must be specially commented.

I recommend the manuscript to be amended.

Author Response

Response to Reviewer 1 Comments

Four polymers have been prepared incorporating an Ala-Phe dipeptide of the LL, LD, DL and DD configuration. All polymer samples have been extensively characterized and then subjected to degradation under two different conditions: as a powder suspended in a phosphate buffer and as a membrane in an aqueous soil suspension. In the both above cases the links with the LL-dipeptide were found to decompose significantly faster than those with the DD-enantiomers.

Thank you for affirmation of our research results, also thank you for your valuable comments. We have carefully considered the following points and made necessary revisions accordingly by following your suggestions. We hope it could meet with your approval.

Point 1: According to stereochemical rules, the LL-polymer must be identical to the DD-analogue with the exception of contacts with living organisms. The degradation experiments of the above four polyamides are exactly the case. I find that this condition is not shown clearly enough in the manuscript. Thus, the fact that the phosphate buffer and the polymer samples have not been previously subjected to sterilization must be clearly spelled out; so that the unavoidable microbial or fungal contamination of the system within the 7 weeks of the experiment presents the only real cause of the difference in the degradation of LL- and DD-enantiomers in the buffer solution. (The pH value of the buffer must be specified).

Response 1: Thank you very much for the good and important advice. We very much agree with your comment. In our experiments, PBS buffer, polymer samples and glassware have not been sterilized beforehand. We have added the important experimental condition of non-sterilization in the manuscript, and the pH of the buffer is also listed. We autoclaved all the materials and degraded the samples in PBS buffer for 7 days. The results confirm that there is no difference in the molecular weight loss rate of polymers with different configurations. As you say, microbial or fungal contamination is the only real cause of the difference in the degradation of LL- and DD-enantiomers.

Point 2: The signs of the specific optical activity of the LL- and DD-polymer solutions MUST BE OPPOSITE, while in the Table 2 the both are shown to be positive. If I assume that this error is a simple misprint, the difference in the absolute values still appears to be exceedingly big: 153 and 115. These values must be re-examined or this point must be specially commented.

Response 2: Thank you very much for this comment. We measured the specific rotation of four polymers with different configurations in DMF solvent, and the results showed that the specific rotations of the polymers were all positive. We believe that the reason for this phenomenon is that the peptide bonds in PAI may agglomerate in DMF solvent, causing a change in the conformation of the polymer, resulting in a change in the rotatory direction and optical rotation. For this reason, we changed the solvent to DMSO to test the optical activity of PAIs. The specific rotations of LL-, LD-, DL- and DD-polymers were obtained as -161.11, +105.33, -108.42 and +159.65, respectively. We have revised it in the manuscript.

Reviewer 2 Report

To determine whether or not the chirality and configuration of the amino acids play an important role in degradation process, the authors synthesized 4 types of polyamide-imides (PAIs) containing dipeptides of LL, DL, LD, and DD configurations. They concluded that the natural L-configuration of amino acids play a critical role in degradability. The purpose of the research is clear and the characterization of the polymer is fine. However, the detail of the degradation process lacks important information.

In line 121, it is written as follows. “The degraded products remained in the supernatant were gained after centrifugation of the above-collected buffer solution”. What is the main chemical species in the supernatant? It is expected that they are oligoamide-imides. And if it is so, how many repeat unit does these species have? The solubility of the oligomer is important in this experiment.

In line 177, which covalent bond is cleaved for the PAI? Isn’t there a possibility of breaking not only the peptide bond between alanine and phenylalanine but also the phenylalanine and ODA?

What is the mechanism of the degradation? It could be caused by bacteria in PBS buffer but if the PBS buffer solution as well as the glassware is autoclaved, then does the degradation process not occur?

In line 125, “The weight losses in percentage for each PAI samples were subsequently recorded as the weight difference between the dried collected PAI samples and the initial ones”. In Figure 2, it is indicated that there are some amount of H2O remained in the sample and it will affect the value of weight losses. How much H2O remained in the sample?

In Table 3, information of molecular weight is provided but the GPC chart should also be provided in supporting information.

Author Response

Response to Reviewer 2 Comments

To determine whether or not the chirality and configuration of the amino acids play an important role in degradation process, the authors synthesized 4 types of polyamide-imides (PAIs) containing dipeptides of LL, DL, LD, and DD configurations. They concluded that the natural L-configuration of amino acids play a critical role in degradability. The purpose of the research is clear and the characterization of the polymer is fine. However, the detail of the degradation process lacks important information.

Thank you for affirmation of our research results, also thank you for your valuable comments. We have carefully considered the following points and made necessary revisions accordingly by following your suggestions. We hope it could meet with your approval.

Point 1: In line 121, it is written as follows. “The degraded products remained in the supernatant were gained after centrifugation of the above-collected buffer solution”. What is the main chemical species in the supernatant? It is expected that they are oligoamide-imides. And if it is so, how many repeat unit does these species have? The solubility of the oligomer is important in this experiment.

Response 1: Thank you very much for this comment. We separated the degraded supernatant and degradation products of PAIs. We also believe that the supernatant contains degraded oligoamide-imide, and want to test its structure and molecular weight. However, the supernatant contains a large amount of phosphate, and we have tried various methods to separate them, but failed to achieve this. This is very important for the interpretation of the degradation mechanism, and we will do a lot of work on this issue in the future.

Point 2: In line 177, which covalent bond is cleaved for the PAI? Isn’t there a possibility of breaking not only the peptide bond between alanine and phenylalanine but also the phenylalanine and ODA?

Response 2: Thank you very much for this comment. We believe that the breakage of peptide bonds in the degradation process of PAI is dominant. According to our previous studies on PAIs containing a single amino acid, the difference in amino acid chirality has only a small effect on the degradation of the polymers. But after introducing peptide bonds, the configuration of amino acids has a great influence on polymers degradation. The imide group between PMDA and alanine, the amide bond between phenylalanine and ODA will also be cleaved, which should not be the main factor in the degradation of PAIs containing dipeptides. In the next work, we will focus on the mechanism of bond cleavage, including the order and degree of bond cleavage. Regarding the cleavage of covalent bonds, we have added in the manuscript.

Point 3: What is the mechanism of the degradation? It could be caused by bacteria in PBS buffer but if the PBS buffer solution as well as the glassware is autoclaved, then does the degradation process not occur?

Response 3: Thank you very much for the good and important advice. We very much agree with your comment. In our experiments, PBS buffer, polymer samples and glassware have not been sterilized beforehand. We have added the important experimental condition of non-sterilization in the manuscript. And according to your suggestion, we autoclaved all the materials and degraded the samples in PBS buffer for 7 days. The molecular weight of the polymer is slightly reduced due to hydrolysis, but there is no difference in the molecular weight loss rate of the polymers with different configurations. This indicates that the differences in the degradability of polymers with different configurations are caused by microorganisms.

Point 4: In line 125, “The weight losses in percentage for each PAI samples were subsequently recorded as the weight difference between the dried collected PAI samples and the initial ones”. In Figure 2, it is indicated that there are some amount of H2O remained in the sample and it will affect the value of weight losses. How much H2O remained in the sample?

Response 4: Thank you very much for this comment. Our samples are all heated and dried, so theoretically should be free of water. In the FTIR spectrum, there are broad peaks at 3000 cm-1-3500 cm-1. We believe that this is due to the cleavage of peptide bonds in PAIs with degradation, leading to the increase of -OH and -NH2 groups, thus producing a broad peak here.

Point 5: In Table 3, information of molecular weight is provided but the GPC chart should also be provided in supporting information.

Response 5: Thank you for this comment. We have provided the GPC diagrams in the supporting material.

Round 2

Reviewer 1 Report

I find the amendments made to the manuscript to be sufficient, and I have no more objections to the acceptance of the modified version.

          Still, I find the data for the optical rotation of the polymer solutions in DMF to be impossible, whereas the data for DMSO which now appear in Table 1 to be reasonable.